



# Influence of atmospheric dynamics during events Secondary Effect of the Antarctic Ozone Hole on southern Brazil

Gabriela Dornelles Bittencourt[1], Damaris Kirsch Pinheiro[1], José Valentin Bageston[2], Hassan Bencherif[3;6], Luis Angelo Steffenel[4] , Lucas Vaz Peres[5]

[1] Federal University of Santa Maria, Santa Maria – RS, Brazil
[2] National Institute for Space Research, Southern Regional Space Research Center, Santa Maria – RS, Brazil
[3] University of Reunion Island, LACy, UMR 8105, Reunion, France
[4] Centre de Recherche en STIC, Université de Reims Champagne-Ardenne, Reims, France.
[5] Federal University of Western Pará, Santarém – PA, Brazil
[6] School of Chemistry and Physics, University of KwaZulu-Natal, Westville, Durban, South Africa

*Correspondence to*: Gabriela Dornelles Bittencourt (gadornellesbittencourt@gmail.com)

**Abstract.** The Antarctic Ozone Hole (AOH) directly influences the Antarctic region where its levels can reach values below 220 UD. The temporary depletion of ozone in Antarctic generally occurs between the beginning and mid-August, during the austral spring, and extends the month of November, where a temporary reduction in ozone content is observed in the Antarctic region. However, masses of ozone-depleted air can break away from the Ozone Hole and reach mid-latitude regions in a phenomenon known as the Secondary Effect of the Antarctic Ozone Hole. The objective of this work is to show how the atmospheric dynamics behaves during the occurrence of this type of event, especially in regions of medium latitudes such as southern Brazil, besides statistical analyzes of the meteorological fields here, or a period of 12 years of observations. For the analysis and identification of the events of influence of the AOH on the southern region of Brazil, data from the total ozone column were used from ground-based and satellite experiments, the Brewer Spectrophotometer (MKIII # 167) and the OMI (Ozone Monitoring Instrument) on the Aura satellite. For the analysis of the stratospheric and tropospheric fields, the ECMWF reanalysis were used. Thus, 37 events of influence of the AOH that reached the southern region of Brazil were identified for the study period (2006-2017), where the events showed that in approximately 70% of the cases they occurred after the passage of frontal systems and / or atmospheric blocks over the southern region of Brazil. In addition, the statistical analyzes showed a strong influence of the jet stream on the mid latitude regions during the events. Of the 37 events identified, 92% occurred with the presence of the subtropical and / or polar jet stream over the study region, possibly explaining the exchange of air masses of ozone-deficient in the UT-LS (Upper Troposphere – Lower Stratosphere) region.

## 1 Introduction

Discovered in 1840 by Christian F. Schonbein, ozone is the most important gas trace constituent of the stratosphere which along with water vapor ($H_2O$) and carbon dioxide ($CO_2$) is responsible for the energy balance of the Earth (SEINFELD AND



PANDIS, 2016). Due to its ability to absorb ultraviolet (UV) radiation, which is more harmful to living beings on Earth (SALBY, 1996 and DOBSON, 1968), it is the most important component of the stratosphere. Most of the atmospheric ozone content (about 90%) is concentrated in the stratosphere between 15 and 35 km altitude (LONDON et al., 1985) in the region known as the Ozone Layer.

The concentration of ozone in a particular region of the Earth is mainly determined by the southern transport of this element
in the stratosphere (GETTELMAN et al., 2011). The explanation for the higher concentration of ozone found in polar rather than equatorial regions (where there is greater production) is precisely a special type of poleward transport known as the Brewer-Dobson circulation, in which air masses are transported quasi-horizontally from the stratospheric tropical reservoir to polar regions (BREWER, 1949; DOBSON, 1968, BENCHERIF et al., 2007; BENCHERIF et al., 2011). The poleward transport of stratospheric ozone is one of the essential factors for the concentration of this atmospheric constituent in a
certain region of the planet (PLOEGER et al., 2012), being much studied from the use of Potential Vorticity, which correlates with the transport of chemical constituents traces such as ozone ($O_3$), nitrous oxide ($N_2O$) and water vapor ($H_2O$) on isentropic surfaces in the lower stratosphere. The potential vorticity acts as a dynamical tracer for large-scale air mass transport, behaving as a material surface where the potential temperature is conserved (HOSKINS et al., 1985). In the lower reaches of the stratosphere the lifetime of $O_3$ molecules is longer and therefore they can be used as a tracer in the study of air
mass flow in the Stratosphere-Troposphere Exchange events (BUKIN et al., 2011).

The first studies with respect to this ozone concentration on Polar Regions showed that during the spring of the Southern Hemisphere there was a massive reduction of the $O_3$ content in this period, being known as Antarctica Ozone Hole (AOH) (CHUBACHI et al., 1984, FARMAN et al., 1985 and SOLOMON et al., 1999).The ozone hole area is defined when there is a region with values below 220 DU, less than two thirds of the historical level (HOFMANN et al., 1997). Nevertheless,
temporary destruction directly influences ozone content in and around the Polar Regions due to the crossing of the polar vortex boundary over these regions, causing drastic reductions in the ozone content and increase of the levels of surface ultraviolet radiation (CASICCIA et al. 2008). However, their effects can affect regions of mid-and low-latitudes, causing temporary decreases in the total columns of ozone.

Poor ozone air masses are released from the interior of the Antarctic polar vortex, the edge of the Ozone Hole, being carried
by the polar filaments on these regions (MARCHAND et al. 2005), in a phenomenon called "Secondary effect of the Antarctic Ozone Hole" causing a temporary fall in ozone content, first observed by Kirchhoff and collaborators (1996) over the South of Brazil. PERES et al. (2014 and 2016) showed the effects of this secondary event on mid-latitude regions such as the southern region of Brazil, where ozone content falls over the region from August to November. Recently, Bittencourt et al. (2018) reported on the second most intense event ever recorded in the southern region of Brazil. According to the latest
WMO reports (2014 and 2018) there is a growth trend between the 1980s and 1990s, stabilizing at high rates since the 2000s, despite indications of declining trends in the Antarctic ozone in recent years (SOLOMON et al., 2016).



Unlike other regions of Brazil, the weather conditions in southern Brazil are strongly influenced by transient meteorological systems (REBOITA et al., 2010). Examples of such systems are cold and hot fronts, which carry strong west winds at high tropospheric levels called jet streams. Moreover, the UT-LS region in southern Brazil seems to be the home place of many

dynamical processes such as stratosphere-troposphere exchanges and isentropic transport between the tropical stratosphere reservoir, polar vortex and mid-latitude. Indeed, understanding the patterns of the UT-LS is important in understanding transport and exchange processes, and the links with tropospheric meteorology (OHRING et al., 2010).

## 2 Data and methodology

### 2.1 Study region and instruments


The region of study was the central region of Rio Grande do Sul, comprising the city of Santa Maria – RS (29.72°S; 53.72°O). This mid-latitude region presents a well distributed precipitation regime throughout the year, approximately 1050 a 1750 mm/year (REBOITA et al., 2010). In this work two instruments were used for the analysis of the total ozone content over the southern region of Brazil for the period of 12 years of data (2006-2017).


The ground surface instrument, the Brewer Spectrophotometer MKIII #167, located in São Martinho da Serra –RS, in the South Space Observatory (SSO), about 30 km from the city of Santa Maria - RS, and also satellite data *Ozone Monitoring Instrument* (OMI) for days when there were no Brewer measures completing the database for the same study region. In the SSO since 2002, the Brewer Spectrophotometer MKIII #167, an automated surface instrument measuring the overall solar

radiation in the Type B Ultraviolet (UVB) band for five wavelengths 306.3; 310.1; 313.5; 316.8; 320.1 nm, where every 0.5 nm determines the spectral distribution of the incident radiation intensity. It allows retrieving the total columns of the following atmospheric gases: ozone ($O_3$), sulfur dioxide ($SO_2$) and nitrogen dioxide ($NO_2$).

The OMI satellite was launched in July 2004 on board the ERS-2 satellite, and continued with the records of the TOMS

satellite that ended its activities in 2005, for TOC and other atmospheric parameters related to ozone chemistry and climate as, $NO_2$, $SO_2$, and can distinguish between types of aerosols, such as smoke, dust and sulfates, and measures the pressure and cloud cover. The earth is observed in 740 bands of wavelength along the satellite route with a band large enough to provide global coverage in 14 orbits (1day). The 13 x 24 km spatial resolution can be expanded to 13 x 13 km to detect and track sources of pollution on an urban scale, having two ultraviolet bands named UV-1 (270 to 314 nm) and UV-2 (306 to 380

nm), with a spectral resolution of 0.45 and 1 nm respectively.

We also used reanalysis data available in (ECMWF/ERA-INTERIM, Daily (2017)) (DEE et, al., 2011), where meteorological fields were prepared for the analysis of the stratospheric and tropospheric dynamics. Due to radiosonde limitations on the study region, the spatial resolution used was 2.5° x 2.5° latitude / longitude, responding well to the


objectives of this work, where a higher resolution is not necessary for further details. For the stratospheric dynamics
       analysis, data of potential vorticity and ozone mixing ratio were used at the potential temperature of 265K and 850K
       (Kelvin). For the analyses of tropospheric dynamics, wind data (components u, v and w), geopotential height and layer
       temperature available from 1000 hPa to 1 hPa, at the pressure level, in addition to pressure data at mean sea level. With these
       data, potential vorticity fields were made for the potential temperature levels of 600 K and 700 K. For tropospheric fields,
sea level pressure and layer thickness between 1000 and 500 hPa, horizontal layer cut showing the jet at 250 hPa and Omega
       at 500 hPa, and a vertical cut of the layer between 1000 and 50 hPa of potential temperature and wind (m/s) for the longitude
       of 54º west.

       The HYSPLIT / NOAA model was used to help identifying the events of influence of the Antarctic Ozone Hole over the
study region (ROLPH et al., 2017). The lagrangian HYSPLIT model is a complete system for calculating simple trajectories
       of air parcels as well as complex transport simulations, chemical transformation and deposition (HYSPLIT, 2017). The
       model assumes that a particle follows the wind flow passively, its trajectory is the integration of the position vector of the
       particle in space and time. In this study, the backward trajectory, where the objective is to show the air mass behavior for
       four days before, and an isentropic vertical velocity model. With this tool it is possible to confirm the events through the
creation of retroactive trajectories showing the way of the Antarctic air masses to the region of interest, available in
       (HYSPLIT, 2017).

**2.2 Identification of AOH influence events**

The identification of the events of influence of the AOH is done first by the analysis of the average daily data of the Total
       Ozone Column (TOC) through the instruments of satellites and ground. In this work, 12 years of satellite data were analyzed,
       with the aim of identifying days where the mean daily value of TOC is less than the climatological average for the month of
       analysis, i.e., the climatological average minus 1.5 of its standard deviation value ($\mu - 1,5\sigma$), where, $\mu$ is a climatological
       average for the month of interest, $\sigma$ is the standard deviation, and the value of -1.5 is the criterion chosen from the normal
frequency distribution tests (WILKS et al., 2006). This criterion was also used by Peres (2016), where it is observed that the
       variations around this value can represent well the influences of the ozone content in the study region.

       After identifying the possible days of influence of the AOH over the study region, the analysis of isentropic surfaces is made,
       where absolute potential vorticity (APV) fields are made. For the analysis of the stratospheric dynamics, we used reanalysis
data available on the Era-Interim / ECMWF platform where Absolute Potential Vorticity (APV) fields were analyzed on
       isentropic surfaces for the potential temperature levels: 600K and 700K. Potential vorticity (PV) is used in studies that
       correlate with chemical constituents such as ozone, water vapor and nitrous oxide on isentropic surfaces (adiabatic surfaces
       where the potential temperature remains constant) in the lower troposphere (SCHOEBERL et al., 1989).





In these cases, the PV acts as a dynamic large-scale air mass tracer and can be used as a horizontal coordinate (HOSKINS et al., 1985). In this way, this type of analysis aims to verify the origin of the air masses, where an APV increase is observed when the air mass originates from higher latitudes (like Antarctica) considering the previous days of the event, or equatorial origin, if a decrease of Absolute Potential Vorticity (SEMANE et al., 2006). Bittencourt et al., (2018) and Bresciani et al., (2018) showed the analysis of an extreme event of influence of the AOH on regions of medium latitudes through the analysis

of the stratospheric dynamics with the fields of PV.

In the analysis of potential vorticity fields the air mass trajectory is observed. When there is an increase in APV we have that the mass of air had polar origin, otherwise if it has equatorial origin it is observed a decrease of the APV. As described above, the APV acts as a dynamic marker for large-scale air masses and, thus, observations are made to identify the

secondary effect of AOH, where reductions in $O_3$ content are observed from intense to moderate (BITTENCOURT et al., 2018, PERES et al., 2016).

### 2.3 Tropospheric analysis

After identifying all the events for the study period (2006-2017), meteorological fields were prepared for the analysis of tropospheric dynamics, which aims to show how the troposphere behaved before, during and after the occurrence of every event of Secondary Effect of the AOH identified in southern Brazil. Peres et al.(2014) showed a case study of 2012 presenting two events of influence of the AOH on the southern region of Brazil, in which the synoptic analysis was done for the region on the day of the event, their results showed that one of the events of influence of the AOH occurred after the

passage of a frontal stationary system, where then the arrival of a high pressure system helped to stabilize the region and also in the advance of the air masses poor in $O_3$ configuring the occurrence of the AOH influence event.

The meteorological data for the construction of the pressure fields at sea level and the layer thickness between 1000 hPa and 500 hPa were obtained by the ECMWF, and the purpose here is to check which synoptic systems were predominating during

the events. A presence of the subtropical jet is intended to be displayed in the field of horizontal winds at 250 hPa and Omega at 500 hPa. In addition, ascending and descending surface movements were identified. Another field analyzed was the vertical cut of the atmosphere at different levels of potential temperature (in Kelvin) and wind (in m/s) for the longitude of 54° west. In this case, the jet stream was present at higher levels of the troposphere, which may aid in air exchanges from the stratosphere to the troposphere (SANTOS, 2016).




## 2.4 Statistical analyzes

The average daily data of the ECMWF for the horizontal components of the wind (zonal u and meridional v) and also the vertical wind velocity (Omega - w) were used for these analyzes., in addition to the temperature, geopotential to the available pressure levels between 1000 and 1 hPa, in addition, the component of PV and $O_3$ to the level of 700 K potential temperature. These data were reduced and organized into matrices with the values of daily averages of each of the variables (temperature, u, v and w, geopotential, PV and $O_3$), in a grid of 2.5° latitude by 2.5° longitude, to the levels previously used.


As the set of data used for the preparation of the stratospheric and tropospheric analysis fields was only for the months of interest in the analysis, from August to November of the 12 years of analysis, the reduction of this dataset allowed the separation of the days of interest (days of the events of influence of the AOH on the southern region of Brazil), and subsequent calculation of the monthly averages.


For stratospheric analyzes, mean fields of all identified events are made for -3 days before the event and +3 days after. For the anomaly analysis of the potential vorticity fields, the following expression was used:

$$\Delta(PV) = PV_{anomaly} = PV_{climatological} - PV_{average\ (2006\text{-}2017)} \tag{1}$$


Where the average of all events for each month was used, and decreased from the month's climatological average. For the tropospheric statistical analyzes, the average fields of the identified events of horizontal cut of the atmosphere were made with the objective of analyzing the behavior of the jet stream at 250 hPa and Omega at 500 hPa.

## 190 3 Results and Discussion

In this work, daily average data of the total ozone column were analyzed from the two instruments described above (Brewer Spectrophotometer MKIII #167 and OMI satellite), comprising a period of 12 years of analysis (2006 to 2017), mainly for the months of August to November (austral spring). Comparing the two instruments, the correlation index found was very

high, i.e., $R^2 = 0,965$, what shows that TOC datasets from the Brewer and OMI experiments are consistent with each other. This is in agreement with the findings reported by Peres et al. (2017). In fact, they using TOC data recorded over the SSO site and showed a good correlation of daily and monthly data measured by the two instruments, for the 1992-2014 periods. In fact, they showed a high degree of correlation between the Brewer, and TOMS ($R^2 = 0.88$) then between the Brewer and OMI ($R^2 = 0.93$). Therefore, due to its good correlation, the use of satellite data is correct when there is a lack in surface

observations, without compromising the results and analyzes for the study region.



After the identification of the mean daily data of the TOC, analysis of the monthly climatology on the SSO of these data, the first step for the identification of AOH side effect events over the study region is the analysis of the climatological average for the reference months and the occurrence of AOH, during the extended austral spring period from August to November.

For this, days are chosen in which falls in the ozone content are observed when the average daily value of the total ozone column is less than the average climatological value of the month minus 1.5 of its standard deviation (μ–1,5σ). Table 1 shows the monthly climatological TOC values with monthly standard-deviation, together with the lower TOC limit, for the extended spring season.

After the confinement of the fall limits presented in table 1, we analyzed 90 days where the TOC value of the day was lower than the limit -1,5σ. From these days, using a methodology described above, a total of 37 events were identified that reached the southern region of Brazil from August to November in the 2006 a 2017. As expected, the identified events occurred mostly during October. This is in agreement with the results found by Peres (2016). To exemplify the analysis developed in this work, we present in the next section a case study that occurred on 18 September 2017 that shows an event of secondary

effect of the AOH on the southern region of Brazil.

### 3.1 Case study: event on September 18, 2017

The event that occurred on 18 Sept. 2017 presented a TOC value, measured by the Brewer Spectrophotometer of 271.5 DU,

representing a decrease of approximately 8.5% in comparison with the climatological average for the month of September reported in Table 1. The observed decrease in TOC could be attributed to isentropic transport in the stratosphere.

Fig. 1 shows the PV fields obtained from ECMWF data at 600K and 700K isentropic level in the stratosphere. One can see from Fig.1 that Chile, Argentina, Uruguay, South of Brazil and Paraguay are under the influence of the passage of

stratospheric air masses characterized by APV values greater than 100. We obtained almost the same PV pattern at the 600K isentropic level. As explained above, PV is a conservative dynamical parameter and transport of air masse takes place on isentropic surfaces (HOSKINS et al., 1985). Therefore, PV distributions could be used to determine the origin of air masses. Since PV values are positive in the north hemisphere and negative in the south hemisphere, for convenience, we refer hereafter to the APV (absolute PV), which is positive regardless of the latitude. In Fig.1 PV values higher than 100 are

associated to air masses of polar origin, what suggests that the observed decrease in the total ozone column at SSO, in South of Brazil, results from the transport of air masses of polar origin with low ozone.

To corroborate this hypothesis, the Lagrangian HYSPLIT model was initialized on 18 SEPT. 2017 at SSO location and run for back-trajectory retrievals in the lower stratosphere (see Fig. 2.a). All the stratospheric back-trajectories show that air

masses observed over SSO in the South of Brazil travelled northward and eastward over the polar region. This confirms the





polar origin of the observed air masses. Moreover, Fig.2.b illustrates the global distribution of TOC recorded by OMI experiment on 18 SEPT. 2018. It shows that transport of polar air is characterized by reduction in TOC distribution extending from the polar region up to mid-latitude region. This well illustrates the side effect of AOH, resulting in a decrease in stratospheric ozone during the event.


After the identification of the secondary effect of the AOH on the southern region of Brazil on 09/18/2017, the tropospheric dynamics analysis, to observe how the troposphere was behaving during the occurrence of this event. The passage of frontal systems over the southern region of Brazil is common at this time of year. Thus, one can observe the passage of a frontal system on 09/14/17 (not shown here), which stabilized the atmosphere on the following days. Already from 09/17/2017, the study region was under the influence of a system of high pressure post - frontal and acted in the region as of 09/20/2017.

Figure 3 presents the analyzed fields for the analysis of tropospheric dynamics. The presence of a post-frontal high pressure system can be observed in the region of Argentina and Uruguay, bringing to these regions and their neighborhoods an air mass poor in $O_3$. For the days before confirmation of the side effect event, the region remained unstable from 9/11/17 until one day before the event, which can be explained by the tapering of the isobars corresponding to a thickness of the most compressed layer, besides the presence of an intense temperature gradient. Already for the day of the event 09/18/2017, the formation of a high-pressure system was set up in the region, who moved away to the ocean already on the following days. Under these circumstances, on the surface, we have the presence of a high pressure system post-frontal near the region of interest, which may have helped to transport this air mass $O_3$ to reach mid-latitude regions as the central region of southern Brazil.

Following in the analysis of tropospheric dynamics, the horizontal fields of wind and temperature present the jet at 250 hPa and Omega at 500 hPa, which aims to show the regions of upward and downward movement of the air masses. The presence of the polar jet stream over the southern region of Brazil, Argentina and Uruguay is observed in figure 3c and 3d. The negative values of Omega at 500 hPa persist throughout the period indicating upward movement at lower levels of the atmosphere. With this, the horizontal cut of the atmosphere showed the presence of the polar jet dominates the region until the day of confirmation of the event. For this reason, the arrival of $O_3$ poor air masses on the region can be associated to the performance of a frontal system that passed over the region days before the confirmation of the event, besides the presence of the jet in higher levels of the atmosphere aiding in the air exchanges of the stratosphere for the troposphere, contributing to the temporary drop in $O_3$ content on 09/18/2017.

The vertical cut of the atmosphere between 1000 and 10 hPa of potential temperature and wind in 54º of longitude, figure 3e and 3f, shows the presence of the polar jet stream, in higher levels of the atmosphere, and also the tapering of the isentropic



near the longitude of 30º South until 09/17/2017, indicating a front ramp, which can assist in exchanging air from the highest
levels to the lowest levels on the day of the event.

**3.2 Statistical analyzes: atmospheric dynamics**

Figure 4 shows the mean field of the 37 AOH influence events identified in this work, where potential vorticity fields were
used for the 700 K isentropic level, for three days before and up to three days after the event. Analyzing Figure 4a and 4b, it
can be observed that for -3 days (-3d) the variation of potential vorticity over the region remains stable, without variation in
the content of APV on the south of Brazil, with APV values between 40 and 60PVU.

Already from 2-days before (-2d) the event, Fig.4c, we can observe a slight increase in APV values over the study region,
mainly between Argentina, Uruguay and South of Brazil with APV values from 60 to 80PVU. From one day before (-1d) the
event, the increase of APV over the study region becomes more important, with APV values between 100 and 140 PVU. For
the days after the event, +1d (Fig.4e) and +2d (Fig.4f), air masses with APV higher than 100PVU bound mid-latitude region
in Chile, Argentina, Uruguay and South of Brazil, with values up to 160 PVU. From the third day after the event, we found a
decrease in APV values similar to the -2d situation (not shown). These results indicate that during the 37 identified
secondary effect events due to the AOH development, low-ozone air masses are transported from polar region to mid-
latitudes and covering a wide region over North of Chile, Argentina, Uruguay and South of Brazil. In average, such low-
ozone event may last and affect that sub-region during at least 4 days. This is agreement with previous works published by
Peres (2016).

Plots on Fig.5 show the averaged monthly distributions of potential vorticity anomalies on the 700K isentropic level,
averaged for August, September, October and November over the study period, 2006-2017. The potential vorticity anomaly
fields show the predominance of positive PV anomalies over the South of Brazil, (values around 35 to 55 PVU). From Fig.5a
it can be observed that for August there is a predominance of positive anomalies on the southern region of Brazil, in
accordance with the number of events identified in this month (7 events, see Table 2) over the SSO site in South of Brazil.
The month of November is the month with the lowest number of low-ozone AOH events identified over the region (5 events,
see Table 2), and also shows the predominance of a positive anomaly on the southern region of Brazil, with potential
vorticity anomalies between 10 and 30 PVU. For the months of September and October, positive PV anomalies were very
evident for the period of 12 years of data.

Significant increases in positive photovoltaic anomalies (between 10 and 50 PVU for September, and from 30 to 60 PVU)
are concomitant and consistent with the large number of low-ozone AOH events recorded during these months (12 events in
September and 13 events in October). Physically, it is possible to confirm the importance of these months for the analysis,





due to the greater number of AOH influencing events that affect southern Brazil, as observed by Bittencourt et al. (2018), which is explained by polar filaments that release from the ozone hole region and then bring $O_3$ to mid-latitude regions.


For a better understanding of the tropospheric dynamics during the 37 events identified, medium fields were made for the horizontal and vertical cuts of the atmosphere. Figure 6 shows the average field for the horizontal cut (jet at 250 hPa and Omega at 500 hPa). In the mean of the 37 AOH influence events identified in this study (Table 2), the presence of the jet stream (subtropical or polar) is observed in practically all identified events, Figure 6 confirms this where the presence of the jet stream is observed mainly on the southern region of Brazil. However, there is a predominance of a center with negative values of Omega at 500 hPa, indicating surface convergence, which explains the majority of events identified after the passage of frontal systems over the southern region of Brazil. Therefore, the importance of the jet stream for the vertical distribution of $O_3$ in the atmosphere, and also in air exchanges from the stratosphere to the troposphere (BUKIN et al., 2011; SANTOS, 2016) on the southern region of Brazil.


Finally, Figure 7 presents the average for the 37 AOH influence events identified on the Southern Brazil of the vertical cut of the atmosphere between 1000 and 30 hPa. Similar to figure 6, the presence of the jet stream with an intense nucleus (~ 45 to 50 m / s) near the latitude and longitude of the study region, besides the presence of a jet near 30 hPa, indicating the probable presence of the polar jet current in the average of the events. However, it is confirmed that the jet stream (subtropical and / or polar, depending on the case) was also present at higher levels of the atmosphere. Therefore, analyzing the average tropospheric dynamics of the 37 events of influence of the AOH on the southern region of Brazil, the presence of the polar jet stream, at higher levels of the atmosphere, as well as the presence of the subtropical jet stream probably explaining the transport of $O_3$-poor air masses from polar regions to mid-latitude regions, like the south of Brazil.

**4 Conclusions**

In this work, we analyzed daily total ozone columns (TOC) measured by the Brewer Spectrophotometer (MKIII #167) operational at the SSO site in the South of Brazil, and by OMI instrument from 2006 to 2017. Analysis of TOC datasets revealed 37 low-ozone events that have occurred and extended during the austral spring period (August-September-October-November) over the SSO site. Moreover, examination of potential vorticity fields in the stratosphere (on the 700K isentropic level) and of back-trajectories obtained by the Lagrangian HYSPLIT model showed that the 37 low-ozone events resulted

from the transport of air masses from polar regions to mid-latitudes, and correspond therefore to the secondary effect of the AOH. In addition to that, it was shown from PV and PV anomalies that the detected events extended over a large region covering North of Chile, Argentina, Uruguay and South of Brazil, and may last and affect that sub-region during at least 4 days. In accordance with the period of development of the AOH and with previous published works, we found that most of



the events took place in September (35%) and October (39%), while 17.6% of them were identified in August and 12.7% in

November.

The analysis of the tropospheric dynamics confirmed the importance of the jet stream as the main synoptic system that assists in the exchange of air masses between the stratosphere and troposphere. Of the 37 events, most (92%) of the cases had the presence of the jet stream (subtropical and / or polar, depending on the surface acting synotic system). In addition, on

the surface, events were identified in 70% of cases after the passage of frontal systems over the southern region of Brazil. The performance of a high pressure system, characterized by the stabilization of the atmosphere with downward movement, helps explain the arrival of ozone-depleting air masses from the Antarctic region to mid-latitude regions.

Regarding the statistical analyzes of the tropospheric fields, confirmation of the importance of the jet stream was obtained.

The vertical cut of the atmosphere showed the presence of the two jet streams (polar and subtropical jet) at higher levels of the atmosphere, besides the current lines converge to regions close to 30º S, southern region of Brazil. The average fields of the 37 events identified in the region, show the presence of the jet stream in relation to the horizontal cut (250 hPa jet) and vertical cut (1000 and 30 hPa).

The results found here highlight the importance of the presence of the jet stream as the main synoptic system that helps in the exchange of masses of ozone-deficient air from the stratosphere to the troposphere. It is evident that the two jet streams (subtropical and polar) act together in this exchange mechanism, possibly being, the "link" between the two atmospheric layers during the occurrence of events of side effect of the AOH on the southern region of Brazil.


*Acknowledgements.* This work is part of the Graduate Program in Meteorology of the Federal University of Santa Maria (UFSM) in cooperation with the Regional Center Space Research (CRS) and National Institute of Space Research (MCTIC/INPE), supported by the Coordination of Improvement of Higher Education Personnel (CAPES). The authors would like to express their thanks to CAPES/COFECUB Program, process nº 88887.130176/2017-01, and National Institute

of Antarctic Science and Technology for Environmental Research, CNPq process n° 574018/2008-5 and FAPERJ process n° E-16/170.023/2008, for the financial support. The authors also thank the data provided by NASA (OMI), ECMWF/Era - Interim for daily average data.







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





| Month | Climatology $O_3$ in DU ($\mu$) | Standard Deviation in DU ($\sigma$) | Limit $-1,5\sigma$ in DU ($\mu-1,5\sigma$) |
|---|---|---|---|
| **August** | 283.7 | 12.9 | 264.3 |
| **September** | 290.7 | 10.1 | 275.5 |
| **October** | 284.4 | 7.2 | 273.6 |
| **November** | 281.3 | 9.7 | 266.7 |

Table 1: Monthly climatological values, their standard deviations and limit $-1.5\sigma$ for August, September, October and November for the South Space Observatory (SSO).

| Event Day | $O_3$Reduction | Event Day | $O_3$Reduction | Event Day | $O_3$Reduction | Event Day | $O_3$Reduction |
|---|---|---|---|---|---|---|---|
| **08/07/2006** | 11.9 % | **10/26/2008** | 6.3 % | **09/14/2012** | 8 % | **08/25/2016** | 11 % |
| **08/23/2006** | 9.2 % | **11/01/2008** | 10.4 % | **09/22/2012** | 4.5 % | **09/05/2016** | 8.6 % |
| **09/19/2006** | 8.7 % | **09/03/2009** | 12.9 % | **10/14/2012** | 11 % | **09/12/2016** | 7.5 % |
| **10/07/2006** | 8.3 % | **09/29/2009** | 7 % | **10/23/2013** | 12.3 % | **10/20/2016** | 22 % |
| **10/15/2006** | 7.6 % | **08/08/2010** | 5 % | **08/10/2014** | 5.4 % | **08/26/2017** | 13 % |
| **11/17/2006** | 11.7 % | **09/08/2010** | 4.4 % | **08/22/2014** | 10.1 % | **09/18/2017** | 8.6 % |
| **09/13/2007** | 5.4 % | **10/13/2010** | 4.6 % | **10/13/2014** | 4.2 % | **11/16/2017** | 9.5 % |
| **10/07/2007** | 8.5 % | **10/22/2010** | 8.6 % | **11/03/2014** | 4 % | | |
| **09/28/2008** | 5.3 % | **10/01/2011** | 4.2 % | **09/22/2015** | 6 % | | |
| **10/12/2008** | 7 % | **10/21/2011** | 4 % | **11/03/2015** | 8.5 % | | |

Table 2: Events of Secondary Effect of the Antarctic Ozone Hole over southern Brazil from 2006 to 2017. Average daily TOC value, percentage of $O_3$ reduction with respect to the climatological average of the month.





Figure 1: Potential Vorticity fields at the 600-K and 700-K isentropic level as derived from ECMWF data successively on (a) and (b) 17 SEPT., (c) and (d) 18 SEPT. and (e) and (f) 19 SEPT. 2017. The black symbol indicates the location of the SSO.




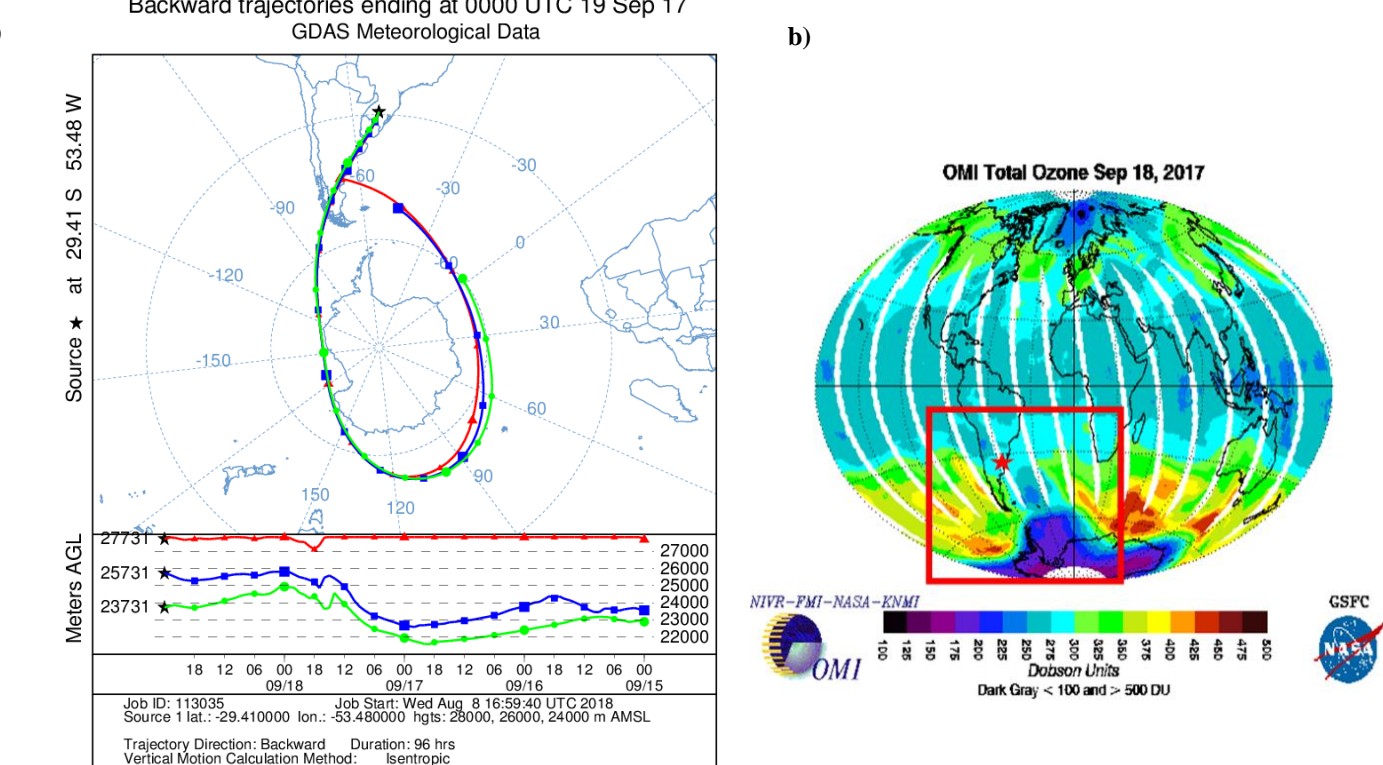

Figure 2: a) Retroactive trajectories as retrieved by model Lagrangian HYSPLIT model, initialized on 09/19/17 00UTC at
SSO location. The back-trajectories were run at 20 km (in red), 24 km (in blue) and 28 km (in green) above ground level; b)
global TOC distribution as recorded by OMI experiment on 18 SEPT. 2017. The red box focus on the low-ozone event, and
the stare symbol indicates the location of the SSO site.





**Sea Level Pressure (hPa) and Thickness (dam)**
a) **09/17/2017**

**Sea Level Pressure (hPa) and Thickness (dam)**
b) **09/17/2017**

**Jet 250 (hPa) and Omega 500 (hPa)**
c) **09/18/2017**

**Jet 250 (hPa) and Omega 500 (hPa)**
d) **09/18/2017**

**Potential Temperature (K) and Wind (m/s)**
e) **09/19/2017 Section lon= -54**

**Potential Temperature (K) and Wind (m/s)**
f) **09/19/2017 Section lon= -54**


Figure 3: a) and b) Pressure fields at medium sea level, c) and d) horizontal cut of the atmosphere and e) and f) vertical cut between 1000 and 50hPa for days 09/17/2017 and 09/18/2017. The red symbol indicates the location of the SSO site.




Figure 4: AveragedAPV maps at the700K isentropic level from 37 APV distributions detected as secondary effect events of AOH: a) -3 days, b) -2 days, c) -1 day, d) day of the event, e) +1 day, f) +2 days.






a) **August Anomaly 2006 – 2017 (700 K)**

b) **September Anomaly 2006 – 2017 (700 K)**

c) **October Anomaly 2006 – 2017 (700 K)**

d) **November Anomaly 2006 – 2017 (700 K)**

Figure 5: Monthly anomaly fields for the period from 2006 to 2017, at the isentropic level of 700 K potential temperature.







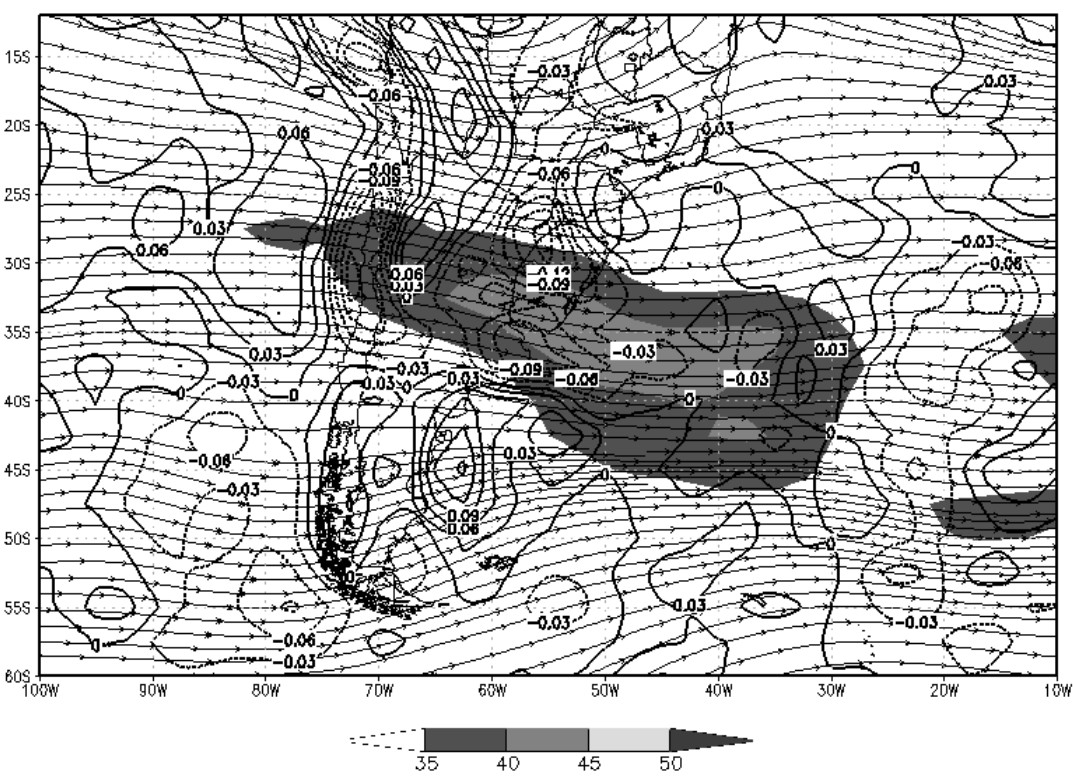

Figure 6: Mean field for the 37 AOH side effect events in the analysis period with jet at 250 hPa (shaded) and Omega at 500 hPa (Omega positive solid lines, Omega negative dotted lines).








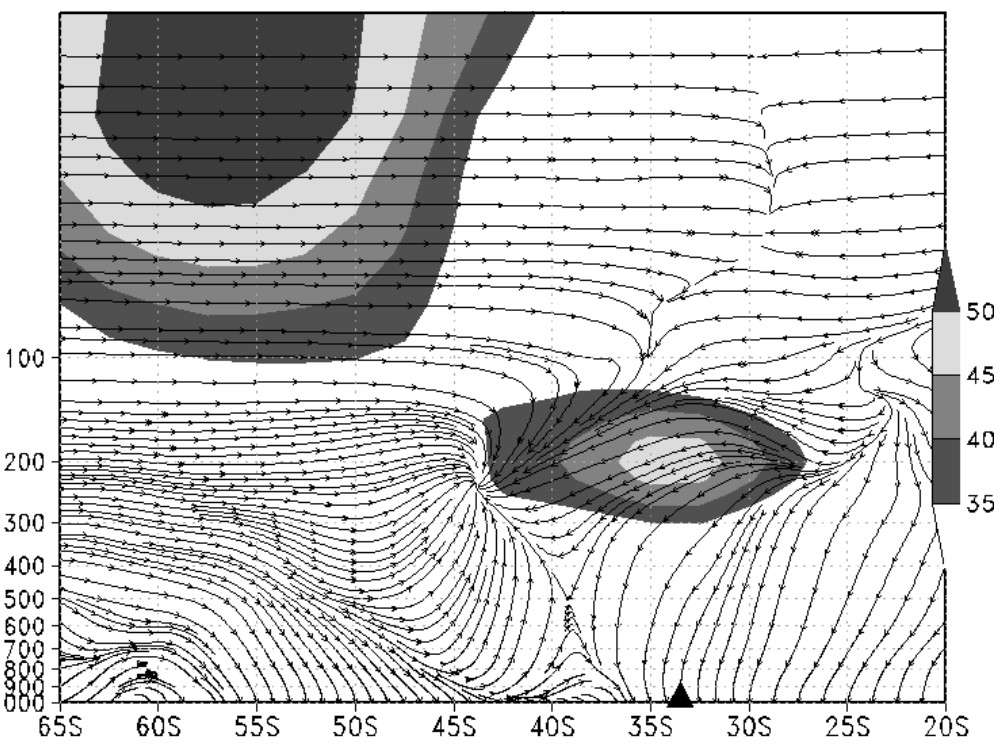

Figure 7: Average for the 37 events of the vertical field between 1000 and 30 hPa, showing the jet current (shaded gray) in m/s.