# Peer review of "Investigation on the behavior of the atmospheric dynamics during occurrences of Ozone Hole's Secondary Effect at Southern Brazil"

_Annales Geophysicae, 2019_

## Referee Comment (RC1) · Anonymous Referee #1 · 17 Sep 2019

The paper study very important phenomenon for the Southern hemisphere. It brings several new aspects but I have found several problems which should be solved before acceptance. 1) Whole troposheric analysis which is supported by Fig. 3 is very confusing. Especially Fig 3 has wrong description for each figures (the same date for both examples). Next I think that it should be read by some meteorologist to improve this part. 2) I would like to ask author why they choose as a case study 18 Sep. 2017 if they can find stronger event during whole period. This should be discussed in more details. 3)In discussion part should be mentioned what is the reason for the rest (you say 92% has jet stream presense) of the events. If this problem is solved I can recommend manuscript for publication.

---

## Author Comment (AC1) · 18 Sep 2019

We thank the Anonymous Referee #1 for fruitful comments and the question/doubts on the paper. All the questions/doubts will be replied point by point just below. NOTE:The new inserts in the paper are in red color.

REFEREE COMMENTS/QUESTIONS AND REPLIES:

1. The total tropospheric analysis, which is supported by Figure 3, is very confusing. Especially, Fig. 3 has an incorrect description for each figure (same date for exple examples). Next, I think it should be read by some meteorologist to improve this part. REPLY: Thank you for considering this part of the work and have suggested clarification, especially about Figure 3. The changes have been made and the text has been rearranged to better understand the new version of the article, as seen on page 8 (Lines 242 to 266) Errors occurred and the dates were really wrong, but now they have been corrected.

2. I would like to ask the author why they chose as a case study on September 18, 2017 if they can find a stronger event throughout the period. This should be discussed in more detail. REPLY: The event was chosen on 09/18/2017 as it was the most recent event identified in our studies. Previous works already identificatied this type of event over the region of interest. For example, in October 2016 there was the second AOH influence event ever recorded in southern Brazil, and it was deeply discussed by Bresciani et al. (2018) and Bittencourt et al. (2018), and in addition to these studies, other works show similar events occurring in that region.This explanation was added on page 7 (lines 215-218).

3. In the part of the discussion, it should be mentioned what is the reason for the rest (you say 92% has jet stream present) of the events. If this issue is resolved, I can recommend the manuscript for publication. REPLY: Thanks for the note, and the corrections have also been changed in the text for better understanding. This explanation was added on page 10 and 11 (lines 333-343).

Please also note the supplement to this comment:
https://www.ann-geophys-discuss.net/angeo-2019-104/angeo-2019-104-AC1-supplement.pdf

**Supplement:**

[revised manuscript text omitted]

**Sea Level Pressure (hPa) and Thickness (dam) 09/17/2017**

**Sea Level Pressure (hPa) and Thickness (dam) 09/18/2017**

**Jet250 (hPa) and Omega500 (hPa) 09/17/2017**

**Jet250 (hPa) and Omega500 (hPa) 09/18/2017**

**Potential temperature (K) and Wind (m/s) 09/17/2017**

**Potential temperature (K) and Wind (m/s) 09/18/2017**

Figure 3: a) and b) Pressure fields at medium sea level, c) and d) horizontal cut of the atmosphere and e) and f) vertical cut between 1000 and 50hPa for days 09/17/2017 and 09/18/2017.The red symbol indicates the location of the SSO site.

[Figure]

Figure 4: Average APV maps at the 700 K isentropic level from 37 APV distributions detected as secondary effect events of AOH: a) -3 days, b) -2 days, c) -1 day, d) day of the event, e) +1 day, f) +2 days.

[Figure]

Figure 5: Monthly anomaly fields for the period from 2006 to 2017, at the isentropic level of 700 K potential temperature.

[Figure]

Figure 6: Mean field for the 37 AOH side effect events in the analysis period with jet at 250 hPa (shaded) and Omega at 500 hPa (Omega positive solid lines, Omega negative dotted lines).

**Average 37 events (2006 - 2017)**
**Vertical Field (m/s)**

[Figure]

Figure 7: Average for the 37 events of the vertical field between 1000 and 30 hPa, showing the jet current (shaded gray) in m/s.

---

## Referee Comment (RC2) · Anonymous Referee #2 · 19 Sep 2019

The article lists some interesting findings regarding the state of the atmosphere during low ozone events at south Brazil, caused by transfer of O3-poor air from the Antarctica. The scientific results presented in the manuscript certainly deserve publication. However, the manuscript has to be improved significantly prior to its publication.

The "Secondary Effect of the Antarctic Ozone hole" refers to low ozone events over mid- and low-latitudes of the southern hemisphere due to the flow of ozone-poor air masses from Antarctica to the mid-latitudes. The authors identify and discus these events, and then describe the changes in atmospheric dynamics throughout their evolution. Finally they show that during these events very strong winds flow at higher (polar jet stream) and lower (sub-tropical jet stream) levels of the atmosphere, which probably explaining the transport of O3-poor air masses from polar regions to mid-latitudes.

The main problem of the manuscript is the language. In several points it was very difficult for me to understand what the authors mean. The authors must try hard to improve the language of the manuscript. Some possible corrections are listed below. Though, more work is necessary. Furthermore, there are several points where the citations either are missing or are not appropriate.

**Analytical comments** are provided below:

**Title:** The title is unclear and confusing and has to change. An alternative title could be "Investigation of the Secondary effect of the ozone hole at Southern Brazil" or something similar.

**P1, l17:** "Antarctica" instead of "Antarctic"

**P1, l22:** "besides … observations." This phrase is very unclear. I recommend re-writing it.

**P1, l23:** define AOH at line 20 before using it here – or use the full phrase.

**P1, l26:** "ECMWF reanalysis products" instead of "ECMWF reanalysis"

**P1, l29:** "analysis" instead of "analyzes". Please correct this error throughout the manuscript.

**P1, l30:** "region of study" instead of "study region"

**P2, l35:** I suppose that the authors mean here, that UV is more harmful than visible radiation. However this information is inaccurate. The biological significance of UV radiation is of course very high, but UV is both beneficial and harmful. Furthermore, the cited literature here does not discuss the biological effects of UV radiation (it is probably at a wrong place?). Since a huge amount of bibliography is available describing the biological effects of UV, I recommend that the authors should search more carefully and add some appropriate references.

**P2, l39:** What is the "southern transport"? Do the authors mean "meridian transport"?

**P2, l51:** "discussing" instead of "with respect to this"

**P2, l52:** Delete "in this period"

**P3, l77-78:** "This … mm/year" Is this information necessary or useful for the study? If no, I suggest removing this sentence.

**P3, l80:** Lines 81 – 87 are written very badly. I suggest trying to re-write more carefully and in a clearer way, and add the appropriate references. For example, I suggest replacing "The … #167" with: "Ground based measurements of the total ozone were performed using the Brewer (type MKIII) with serial 167, now on referred as MKIII #167" or something similar. Furthermore, either discuss the reliability and uncertainties in the total ozone measurements from Brewer and OMI, or at least provide the appropriate references.

**P3, l85-87:** Brewer has two operational modes. It can either measure – nearly simultaneously – the irradiance at the referred wavelengths (306.3, …, 320.1 nm) or scan the solar spectrum with a step of 0.5 nm in a particular wavelength range (for MKIII Brewers it is usually 290 – 363 nm). I am also pretty sure that $NO_2$ cannot be retrieved from spectral measurements in the UV-B region as authors state (although the MKIII type Brewers such as the one used here also provide measurements in the UV-A region where it is possible to retrieve $NO_2$). Please investigate the relative bibliography and add more accurate information, as well as the appropriate references.

**P3, l89:** OMI is not a satellite. It is an instrument on board on Aura satellite. TOMS (total ozone monitoring instrument) is also not a satellite, but a satellite instrument. Please be more careful and add the appropriate references.

**P3, l89-95:** Although authors discus the retrieval of many different products from OMI (without however citing the appropriate literature), they do not provide any information or reference about the retrieval of total ozone. Since there are many studies regarding the validation of the OMI total ozone product, I also suggest adding some relative discussion in order to highlight the reliability of the total ozone measured by OMI. In all cases, please add the appropriate references. Finally, please specify if TOC is the total ozone column.

**P4, l103:** Please add "were used" after "sea level".

**P4, l114:** Please add "were used" after "velocity model

**P4, l124:** "$\sigma$ is"

**P4, l128-129:** "After … made". This sentence is unclear. Please re-write it.

**P6, l171:** geopotential height?

**P6, l186:** "subtracted" instead of "decreased"?

**P6, l192:** Again, OMI is not a satellite

**P6, l199-200:** I do not agree that a strong correlation is enough in order to allow merging the ground-based and satellite datasets. The authors should also discuss the average, as well as the maximum differences between the two datasets. If for example there is –even a small - offset this would directly introduce a bias in the results of the analysis. Furthermore, if there are differences of 5-10 DU between the satellite and ground-based measurements (even for a very limited number of days), then how the authors know that they are not affecting the results? I suggest discussing the above issues here in order to prove that the merging does not affect importantly the results of the present study.

**P7, l229:** Delete "(absolute PV)"

**P8, l246:** Delete "for the analysis of tropospheric dynamics"

**P8, l252:** Replace "who" with "which"

**Figure 4:** The PV, and not the absolute PV is resented in the figure. Please correct the caption.

**Figure 5:** Define on the figure caption that the anomalies of the PV are presented here.

**P9, l300:** What is the meaning of "photovoltaic" here? Is it a typo?

---

## Author Comment (AC2) · 3 Oct 2019

We thank the Anonymous Referee #2 for fruitful comments and the question/doubts on the paper. All the questions/doubts will be replied point by point just below. NOTE: The new inserts in the paper are in blue color. NOTE2: All corrections can be checked in the attached file.

REFEREE COMMENTS/QUESTIONS AND REPLIES:

Title: The title is unclear and confusing and has to change. An alternative title could be "Investigation of the Secondary effect of the ozone hole at Southern Brazil" or some-

thing similar. Answer: We understand that the referee could found the title a little confused, but we are analyzing the influence of the atmospheric dynamics (what are the main characteristics/behavior of the atmospheric) when it occurs the Secondary Effect of the Ozone Hole in South Brazil. So, we tried to re-phrase the title in order to be clearer: 'Investigation on the behavior of the atmospheric dynamics during occurrences of Ozone Hole's Secondary Effect at Southern Brazil'

P1, l17: "Antarctica" instead of "Antarctic" Answer: The suggestion was accepted.

P1, l22: "besides . . . observations". This phrase is very unclear. I recommend rewriting it. Answer: The suggestion was accepted, and can be seen on page 1 line 20-22.

P1, l23: define AOH at line 20 before using it here – or use the full phrase. Answer: The definition of the term was made at the beginning of the abstract, line 16.

P1, l26: "ECMWF reanalysis products" instead of "ECMWF reanalysis". Answer: The suggestion was accepted.

P1, l29: "analysis" instead of "analyzes". Please correct this error throughout the manuscript. Answer: The suggestion was accepted.

P1, l30: "region of study" instead of "study region" Answer: The suggestion was accepted.

P2, l35: I suppose that the authors mean here, that UV is more harmful than visible radiation. However this information is inaccurate. The biological significance of UV radiations is of course very high, bur UV is both beneficial and harmful. Furthermore, the cited literature here does not discuss the biological effects of UV radiation (it is probably at a wrong place?). Since a huge amount of bibliography is available describing the biological effects of UV, I recommend that the authors should search more carefully and add some appropriate references. Answer: Indeed the text was out of context and does not justify the importance of the study in relation to ultraviolet radiation. The text

has been corrected and can now be viewed on page 3 on lines 57-60.

P2, l39: What is the "southern transport"? Do the authors mean "meridian transport"? Answer: The correction has been made, and is in blue color, on page 2 line 39.

P2, l51: "discussing" instead of "with respect to this" Answer: The suggestion was accepted.

P2, l52: Delete "in this period" Answer: The suggestion was accepted.

P3, l77-78: "This. . . mm/year" Is this information necessary or useful for the study? If no, I suggest removing this sentence. Answer: The information would only be to complement the description of the region of study, but it has already been taken from the text and is not very related to the subject.

P3, l80: Lines 81-87 are written very badly. I suggest trying to re-write more carefully and in a clearer way, and add the appropriate references. For examples, I suggest replacing "The. . . #167" with: "Ground based measurements of the total ozone were performed using the Brewer (type MKIII) with serial 167, now on referred as MKIII #167" or something similar. Furthermore, either discuss the reliability and uncertainties in the total ozone measurements from brewer and OMI, or at least provide the appropriate references. Answer: Corrections have been made and text has been fixed, now on pages 2 and 3 on lines 87 – 110.

P3, l85-87: Brewer has two operational modes. It can either measure – nearly simulta-neously – the irradiance at the referred wavelengths (306.3, . . ., 320.1 nm) or scan the solar spectrum with a step of 0.5 nm in a particular wavelength range for MKIII Brewers it is usually 290 – 363 nm). I am also pretty sure that NO2 cannot be retrieved from spectral measurements in the UV-B region as author's state (although the MKIII type Brewers such as the one used here also provide measurements in the UV-A region where it is possible to retrieve NO2). Please investigate the relative bibliography and add more accurate information, as well as the appropriate references. Answer: Cor-

rections have been made and text has been fixed, now on pages 2 and 3 on lines 87 – 110.

P3, l89: OMI is not a satellite. It is an instrument on board on Aura satellite. TOMS (total ozone monitoring instrument) is also not a satellite, but a satellite instrument. Please be more careful and add the appropriate references. Answer: The correction has been made.

P3, l89-95: Although authors discus the retrieval of many different products from OMI (without however citing the appropriate literature), they do not provide any information or reference about the retrieval of total ozone. Since there are many studies regarding the validation of the OMI total ozone product, I also suggest adding some relative discussion in order to highlight the reliability of the total ozone measured by OMI. In all cases, please add the appropriate references. Finally, please specify if TOC is the total ozone column. Answer: The discussion about the OMI data has been corrected, and more references have been added regarding the data used, corrections are on page 8, lines 223 – 231. Total ozone column was defined on page 5, lines 147-148.

P4, l103: Please add "were used" after "sea level". Answer: The correction has been made.

P4, l114: Please add "were used" after "velocity model". Answer: The correction has been made.

P4, l124: "ïĄş is" Answer: The correction has been made.

P4, l128-129: "After . . . made". This sentence is unclear. Please re-write it. Answer: The sentence has been rewritten and is now on the page 5 line 135.

P6, l171: geopotential height? Answer: The geopotential height represents the altitude above sea level at which a certain pressure level, for example here 500 hPa. Thus, geopotential height data are used for tropospheric analysis.

P6, l186: "subtracted" instead of "decreased"? Answer: The correction has been made.

P6, l192: Again, OMI is not a satellite. Answer: The correction has been made.

P6, l199-200: I do not agree that a strong correlation is enough in order to allow merging the ground-based and satellite datasets. The authors should also discuss the average as well as the maximum differences between the two datasets. If for example there is –even a small – offset this would directly introduce a bias in the results of the analysis. Furthermore, if there are differences of 5-10 DU between the satellite and ground-based measurements (even for a very limited number of days), then how the authors know that they are not affecting the results? I suggest discussing the above issues here in order to prove that the merging does not affect importantly the results of the present study. Answer: The correction has been made, available on page 8, lines 223 – 231.

P7, l229: Delete "(absolute PV)". Answer: The correction has been made.

P8, l246: Delete "for the analysis of the tropospheric dynamics" Answer: This sentence has already been corrected in the previous version for Reviewer 1.

P8, l252: Replace "who" with "which" Answer: This sentence has already been corrected in the previous version for Reviewer 1.

Figure 4: The PV, and not the absolute PV is presented in the figure. Please correct the caption. Answer: The caption has been corrected.

Figure 5: Define on the figure caption that the anomalies of the PV are presented here. Answer: The caption has been corrected.

P9, l300: What is the meaning of "photovoltaic" here? Is it a typo? Answer: The correction has already been made. The word was wrong.

Please also note the supplement to this comment:
https://www.ann-geophys-discuss.net/angeo-2019-104/angeo-2019-104-AC2-supplement.pdf

[Figure]

**Supplement:**

[revised manuscript text omitted]

---

## Editor Comment (EC1) · Igo Paulino (Editor) · 6 Oct 2019

Dear Dr. Bittencourt at el.,

Thank you for revise the manuscript according to the reviewers' suggestions. Before sending you manuscript to the production, I would recommend you to revise few points listed below (I have used as reference the file angeo-2019-104-AC2- supplement.pdf):

In the introduction, it is not clear the objectives of the present work and what is the new findings from the authors to the knowledge on the dynamics of the ozone holes.

Line 35: In my opinion, this statement must be re-written. I agree with authors about

the importance of the ozone to the stratosphere and atmosphere, but I would not write it as "the most important constituent" because It depends on the objective of study.

Line 54: delete "in and".

Line 60: mid- and low-latitudes

Lines 65-66: Effects of this secondary event on middle latitude regions such as the southern region of Brazil, where ozone content falls over the region from August to November have been published elsewhere (Peres et al., 2014,2015).

Please, write the meaning of all acronyms as soon as they apear in the manuscript, e.g., SSO, MKII, MKIV, SSO, WMO, UT-LS (described only in the abstract), etc.

Line 269: "...For the day of the event September 18, 2018, it was..."

Line 310: "November is the month.."

Line 345: Please, revise: "It was shown from PV and PV anomalies..." It is confuse.

Why did not you overplot the Jet stream and Omega on the map in Figure 6?

Best regards,

Igo

---

## Author Response (AR1)

We thank Editor Igo Paulino for the suggestions and questions in the paper. All questions / concerns and suggestions will be answered point by point below.

**NOTE: New inserts in the paper are blue.**

EDITOR'S COMMENTS / QUESTIONS AND ANSWERS:

**In the introduction, it is not clear the objectives of the present work and what is the new findings from the authors to the knowledge on the dynamics of the ozone holes.**

Answer: The text has been rearranged and the main purpose of this work has been added. See page 3, line 79.

**Line 35: In my opinion, this statement must be re-written. I agree with authors about the importance of the ozone to the stratosphere and atmosphere, but I would not write it as "the most important constituent" because It depends on the objective of study.**
Answer: The whole sentence has been reworded for a better understanding. See page 2, line 34-35.

**Line 54: delete "in and".**
Answer: The suggestion was accepted, and corrections were made. (page 2, line 57)

**Line 60: mid- and low-latitudes**
Answer: The suggestion was accepted, and corrections were made. (page 2, line 62)

**Lines 65-66: Effects of this secondary event on middle latitude regions such as the southern region of Brazil, where ozone content falls over the region from August to November have been published elsewhere (Peres et al., 2014,2015).**
Answer: All references to this were mentioned in the paper throughout the text. (page 2, line 65-66)

**Please, write the meaning of all acronyms as soon as they apear in the manuscript, e.g., SSO, MKII, MKIV, SSO, WMO, UT-LS (described only in the abstract), etc.**
Answer: Actually, WMO is at the line 68. It was added the meaning of this acronym. The meaning of SSO is at the line 88-89, just before "(SSO/INPE)". The meaning of UT-LS was properly added at line 72 (pg 3).

**Line 269: "…For the day of the event September 18, 2018, it was…"**
Answer: All the sentence was rephrased. Thanks for had noted this mistake. (page 9, line 274)

**Line 310: "November is the month.."**
Answer: The sentence was changed in a proper way. (page 10, line 317)

**Line 345: Please, revise: "It was shown from PV and PV anomalies…" It is confuse.**
Answer: The sentence was changed in a proper way. (page 11, line 355)

**Why did not you overplot the Jet stream and Omega on the map in Figure 6?**

Answer: Figure 6 shows the jet field at 250 hPa represented by shading in the figure, and omega is in positive and negative values, solid lines representing omega with positive values and dotted line omega with negative values.